# Change of Gut Microbiota in PRRSV-Resistant Pigs and PRRSV-Susceptible Pigs from Tongcheng Pigs and Large White Pigs Crossed Population upon PRRSV Infection

**DOI:** 10.3390/ani12121504

**Published:** 2022-06-09

**Authors:** Tengfei Wang, Kaifeng Guan, Qiuju Su, Xiaotong Wang, Zengqiang Yan, Kailin Kuang, Yuan Wang, Qingde Zhang, Xiang Zhou, Bang Liu

**Affiliations:** 1Key Laboratory of Agricultural Animal Genetics, Breeding, and Reproduction of Ministry of Education, Huazhong Agricultural University, Wuhan 430070, China; wangtengfei@webmail.hzau.edu.cn (T.W.); guankf@webmail.hzau.edu.cn (K.G.); suqiuju@webmail.hzau.edu.cn (Q.S.); wangxiaotong@webmail.hzau.edu.cn (X.W.); 1113171404@webmail.hzau.edu.cn (Z.Y.); kuangkl@webmail.hzau.edu.cn (K.K.); 15927387153@163.com (Y.W.); 2Laboratory Animal Centre, College of Veterinary Medicine, Huazhong Agricultural University, Wuhan 430070, China; qdzhang@mail.hzau.edu.cn; 3The Cooperative Innovation Center for Sustainable Pig Production, Wuhan 430070, China; 4The Engineering Technology Research Center of Hubei Province Local Pig Breed Improvement, Wuhan 430070, China; 5Hongshan Laboratory, Huazhong Agricultural University, Wuhan 430070, China

**Keywords:** PRRSV, 16S RNA, gut microbiota, resistant pigs, susceptible pigs

## Abstract

**Simple Summary:**

The gut microbiota could directly induce immune responses and affect the health of the host. In this study, we assessed changes in the gut microbiota of resistant segregated phenotypic pigs under Porcine Reproductive and Respiratory Syndrome Virus exposure. The results showed that the resistance of pigs was related to the composition of gut microbiota. The quantity and relative abundance of probiotics in resistant individuals positively affected host immunity and growth performance, whereas high levels of pathogenic bacteria in susceptible individuals were associated with poorer clinical outcomes. The results of this study suggest that gut microbiota may serve as an effective probiotic resource to provide new methods for PRRS prevention and treatment.

**Abstract:**

Porcine Reproductive and Respiratory Syndrome (PRRS) is one of the serious infectious diseases that threatens the swine industry. Increasing evidence shows that gut microbiota plays an important role in regulating host immune responses to PRRS virus (PRRSV). The aim of this study was to investigate gut microbiota difference between PRRSV-resistant pigs and PRRSV-suspectable pigs derived from a Tongcheng pigs and Large White pigs crossed population. PRRSV infection induces an increase in the abundance and diversity of gut microbiota. Correlation analysis showed that 36 genera were correlated with viral loads or weight gain after PRRSV infection. *Prevotellaceae-NK3B31-group*, *Christensenellaceae-R7-group*, and *Parabacteroides* were highly correlated with both viral load and weight gain. Notably, the diversity and abundance of beneficial bacteria such as *Prevotellaceae-NK3B31-group* was high in resistant pigs, and the diversity and abundance of pathogenic bacteria such as *Campylobacter* and *Desulfovibrio* were high in susceptible pigs. Gut microbiota were significantly associated with immune function and growth performance, suggesting that these genera might be related to viremia, clinical symptoms, and disease resistance. Altogether, this study revealed the correlation of gut microbiota with PRRSV infection and gut microbiota interventions may provide an effective prevention against PRRSV infection.

## 1. Introduction

Porcine Reproductive and Respiratory Syndrome (PRRS) is a major viral infectious disease for the global swine industry, and it mainly causes reproductive disorders in sows, respiratory disease in piglets, and lowers semen quality in boars [1,2,3]. The clinical symptoms mainly include fever, tachypnea, dyspnea, diarrhea, and slow growth, accompanied by secondary bacterial infections, and the infected piglets have a high mortality rate [4,5]. PRRS is caused by porcine reproductive and respiratory syndrome virus (PRRSV), and PRRSV was first isolated as European type I in 1991, and North American type II in 1992, respectively [6,7]. PRRSV is a single- and positive-stranded RNA virus that belongs to the order Nidoviridae, the family Arteriviridae, and the genus Arterivirus [8]. The PRRSV genome is highly susceptible to mutation, and the highly virulent strain was isolated from 2006, indicating a strong heterogeneity among PRRSV strains [9,10]. Recent studies have shown that genetic differences play an important role in different immune response to PRRSV [11,12,13]. After PRRSV infection, Duroc pigs have a higher viral load, severer lung lesions, and a lower average daily weight gain than Meishan pigs [12]. Our previous study has reported that Tongcheng (TC) pigs showed milder clinical symptoms, less lung pathological damage, and a lower serum viral load than Large White (LW) pigs in response to PRRSV infection [14]. These studies indicated that PRRSV resistance is closely related to host genetics. In addition, PRRS disease resistance is also affected by the environment [15], especially the endogenous environment formed by the gut microbiota that interacts genetically with the host.

The mammalian gut is inhabited by about 1000 species of microorganisms, mainly bacteria, called the gut microbiota. The gut microbiota mediates many different forms of responses such as the intestinal barrier, nutrition, metabolism, and immunity [16]. The gut is the distal tissue of the respiratory system, and its immune function is mediated by the gut microbiota, thus, the gut plays an important role in respiratory diseases. Recent studies in sepsis and acute respiratory distress syndrome (ARDS) suggested that certain gut microbes might increasingly translocate across the bowel wall and even enter the lung [17]. Related studies have proposed the concept of the lung–gut axis [18,19], which is an interaction system mediated and regulated by the gut microbiota. Budden et al. (2017) and Niederwerder (2017) found a close link between gut microbiota and respiratory infections, and they also reported the effect of gut microbes on lung disease [20,21]. The reduced gut microbial diversity in pigs infected with PRRSV causes an earlier and stronger dysbiosis of gut microbiota and more losses of anaerobic commensal bacteria such as *Roseburia*, *Anaerostipes*, *Butyricicoccus*, and *Prevotella*, which may be significantly associated with the severity of viremia, clinical symptoms, pulmonary lesions, and immune responses [22]. Similar results have been reported in the study of the co-infection with PRRSV and Porcine Circovirus type 2 (PCV2) [23]. In addition, PRRSV can significantly affect the function of the gastrointestinal tract, thus inhibiting the nutrient absorption in the intestine, and eventually suppressing weight gain. The nutrient metabolism is closely related to the composition of the gastrointestinal microbiota, and the intestinal mucosal barrier establishment and intestinal immune function and structure are modulated through various pathways such as energy conversion [24] and nutrient absorption [25]. *Campylobacter* and *Clostridium* have been reported to possibly be associated with diarrhea since they exhibit a higher relative abundance after PRRSV infection [26]. After PRRSV and PCV2 co-infection, pigs with the best clinical outcome display a higher microbial diversity, a higher mean weekly weight gain, less interstitial pneumonia, a lower viremia level, and a lower serum bacterial load [27]. Notably, the bacteria can participate in the disease resistance process by directly inducing an immune response in pigs [28]. After 42-days of infection, the morbidity (12.5%) and mortality (20%) are significantly lower in the FMT (fecal microbiota transplantation) group than in the control group, and the clinical signs are milder when fecal bacteria from healthy sows are transplanted to tested pigs co-infected with PRRSV and PCV-2 [29].

This studyis aim to analyze the correlation between PRRSV infection and gut microbiota, which helps to provide effective anti-infective fecal agents. The Tongcheng (TC) pigs and Large White (LW) pigs crossed population has been bred since 2008, providing a resource population to study disease resistance in PRRS. Recent studies have revealed the population showed resistant phenotype stratification after PRRSV infection [30,31].

In order to investigate gut microbiota associated with host resistance to PRRSV infection, thirty-five pigs from the PRRSV-resistant TC) pigs and PRRSV-susceptible LW crossed population were infected with PRRSV to explore how gut microbiota affects PRRSV pathogenesis. We aimed to screen for specific microorganisms associated with PRRSV disease resistance based on the trait characteristics of the crossed population. Our results provide new insight into the role of gut microbiota in the host resistance to PRRSV.

## 2. Materials and Methods

### 2.1. Animals Experiment and Sample Collection

Thirty-five 5-week-old healthy piglets from LW pigs and TC pigs crossed population were selected from Yunzhi ecological farm (Tongcheng, Hubei, China). All healthy piglets were selected and housed in one environmentally controlled room at the Huazhong University Large Animal Research Center (Wuhan, Hubei, China). The room was chemically disinfected, cleaned with a high heat pressure washer and gas decontaminated with vaporized hydrogen peroxide prior to use. During experiment period, animals had unrestricted access to food and water with daily health surveillance. BeforePRRSV infection, the pigs were fed for two weeks for environmental adaptation and tested negative for PCV2, PRV, and PRRSV nucleic acids and antibodies.

The 35 pigs were inoculated with 3 mL viral suspension (2 mL intranasally and 1 mL intramuscularly) PRRSV WuH3 strain at a viral dose of 10^5.0^ TCID_50_/_mL_. The body temperatures and clinical symptoms of each pig were recorded daily during PRRSV infection. Blood sampling, weight, and viral RNA detection measurement were conducted as previously described The body weight and venous blood was taken on an empty stomach at 0, 4, 7, 11, 14, 21, 28, and 35 days post-infection (dpi). Total RNA was extracted from blood serum of all experimental pigs and measured by NanoDrop2000 (Thermo Scientific, Waltham, MA, USA). Total RNA for each sample was reverse transcribed using Prime Script RT reagent Kit With gDNA Eraser (TaKaRa, Osaka, Japan). The primer of absolute qRT-PCR was special to PRRSV ORF7 (Forward: 5′-TCAGCTGTGCCAAATGCTGG-3′, Reverse: 5′-AAATGGGGCTTCTCCGGGTTTT-3′, probe: 5′-FAM-TCCCGGTCCCTTGCCTCTGGA-TAMRA-3′). The plasmid pMD18-T carrying PRRSV ORF7 was proportionally diluted into six different concentrations to draw the standard curve. Absolute quantitative RT-PCR was performed with the template of each cDNA sample. The viral load of each sample was calculated by the standard curve equation [14]. Fecal samples were collected from all 35 pigs following aseptic procedures at 0 and 7 dpi and stored at −80 °C until extraction of microbial genomic DNA.

### 2.2. DNA Extraction and 16S rRNA Gene Sequencing

Total DNA was extracted from the fecal samples with QIAamp DNA Stool Mini Kit (QIAGEN, Dusseldorf, Germany) according to the manufacturer′s protocol. The final DNA concentration and purity were detected by NanoDrop 2000 (Thermo Scientific, Waltham, MA, USA). The 16S rRNA V3-V4 region of the eukaryotic ribosomal RNA gene was amplified by PCR (Polymerase chain reaction) using primer pair 341F: CCTACGGGNGGCWGCAG and 806R: GGACTACHVGCGTAAT. PCR reactions were performed in 50 μL mixture system containing 5 μL of 10 × KOD Buffer, 1.5 μL of each primer, 5 μL of 2.5 mM dNTPs, 1 μL of KOD Polymerase, and 100 ng of template DNA in triplicates. PCR products were analyzed on 2% agarose gel electrophoresis and purified with the AxyPrep DNA Gel Extraction Kit (Axygen Biosciences, Union City, CA, USA) according to the manufacturer′s instructions. Then, the purified PCR products were quantified using QuantiFluorTM-ST (Promega, Madison, WI, USA), and pooled in equimolar amounts and paired-end sequenced on an Illumina platform according to the standard protocols.

### 2.3. Sequencing Data Analysis

After data cleaning, chimeric tags were removed to obtain effective tags by UCHIME software [32]. The UPARSE software clustered all the effective tag sequences to obtain operational taxonomic units (OTUs) at 97% sequence similarity [33]. Then the absolute abundance and relative information of tags of OTUs were calculated for each sample. The bacteria were categorized according to representative sequences by a naive Bayesian model using RDP classifier (Version 2.2) [34] based on SILVA Database [35].

### 2.4. Statistical Analysis

The Chao1 index and Shannon index were used to examine the abundance and diversity, respectively. The rarefaction curves were used to validate sequencing results. The indices (Shannon and Simpson) and rarefaction curves were conducted using QIIME [36]. The gut microbiota of 35 piglets at 0 dpi and 7 dpi were analyzed by principal component analysis (PcoA) based on unweighted distance. The abundances of different species of bacteria were analyzed by a Perl script and visualized by SVG, and the relative abundance of bacterial phyla and genera at the different infection time points was compared through Wilcoxon rank-sum test (Benjamini Hochberg FDR) [37]. Spearman′s correlation analyses were performed to reveal potential correlation among the relative abundance of fecal bacterial genera, weight gain, and viral load, and Student′s *t*-test was used to analyze the differences in microbiota between resistant and susceptible pigs. Tax4Fun was used to analyze the 16S rRNA sequences from the SILVA database to obtain OTU′s species annotation and abundance information, based on which KEGG pathway enrichment analyses were performed [38]. All the analyses were conducted by R package.

## 3. Results

### 3.1. PRRSV Infection Induces Increased in Abundance and Diversity of Gut Microbiota

A total of 70 fecal samples were collected and 16S rRNA V3-V4 sequencing data were analyzed. The Alpha diversity analysis was used to examine the species abundance and diversity of the gut microbial community in the samples. The species rarefaction curve (Figure 1A) and Shannon rarefaction curve (Appendix A) exhibited first an increase and then a steadiness trend with an increasing sequencing depth, indicating that all species in the sample were covered by 16S rRNA gene sequencing. The rank abundance curve exhibited a desirable horizontal curve width and vertical curve smoothness, suggesting that the sequencing data had a good category richness and uniformity (Appendix A). The Shannon index showed a significant higher diversity of gut microbiota at 7 dpi than at 0 dpi, but no significant difference in abundance reflected by the Chao1 index was observed between 7 dpi and 0 dpi (*p* = 0.497) (Figure 1B,C). The PCoA showed two obvious gut microbiota clusters, indicating that the composition of the gut microbiota changed dramatically from 0 dpi to 7 dpi (Figure 1D). At 7 dpi, the top 10 dominant phyla were *Bacteroidetes*, *Firmicutes*, *Spirochaetae*, *Proteobacteria*, *Chamydiae*, *Planctomycetes*, *Euryarchaeota*, *Verrucomicrobia*, *Cyanobacteria*, and *Fibrobacteres* with a relative abundance above two percent, and the phyla with significant difference in the relative abundance between 7 dpi and 0 dpi were *Chlamydia*, *Proteobacteria*, and *Verrucomicrobia* (Figure 1E).

### 3.2. Key Gut Microbiota Associated with PRRS Resistance

As shown in Appendix A, the viral load and weight gain of pigs changed greatly during PRRSV infection. PRRSV began to replicate in the host, and the viral load of all pigs was rapidly increased from 4 dpi and reached the peak at 7 dpi with a value of about 8.29 log_10_ copies/mL. The pig feed intake was reduced, and weight gain was slowed down from 0 dpi to 4 dpi. From 4 to 14 dpi, pigs exhibited a poor appetite and a negative weight gain. After 14 dpi, the surviving pigs displayed a gradual increase in weight, and after 21 dpi, the weight increase became dramatic. Pigs showed some symptoms such as coughing, abdominal breathing, convulsions, diarrhea, and cyanosis of the skin from 3 dpi, appearing dead individuals (Appendix A).

To establish a potential link between changes in the gut microbiota and PRRSV resistance, we investigated the relationship between gut microbiota, viral load, and weight gain (Appendix A). Spearman′s correlation analysis showed that viral load was positively correlated with *Campylobacter* (ρ = 0.8), *Ruminococcaceae-UCG−002* (ρ = 0.71), *Ruminococcaceae-NK4A214-group* (ρ = 0.64), *Ruminococcaceae-UCG-010* (ρ = 0.57), *Chlamydia* (ρ = 0.53), *Parabacteroides* (ρ = 0.44), *dgA11-gut-group* (ρ = 0.57), *Christensenellaceae-R.7-group* (ρ = 0.5), *Anaerotruncus* (ρ = 0.42), *Lachnospiraceae**-xpb1014* (ρ = 0.5), *Mitsuokella* (ρ = 0.41), *Alloprevotella* (ρ = 0.32), *Selenomonas* (ρ = 0.32), *Desulfovibrio*(ρ = 0.29), *Bacteroides* (ρ = 0.25), and *Coprostanoligenes-group* (ρ = 0.24), but negatively correlated with *Phascolarctobacterium* (ρ = −0.71), *Lachnospiraceae-NK4A136_group* (ρ = −0.7), *Anaerovibrio* (ρ = −0.48), *coprococcus−1* (ρ = −0.45), *Prevotellaceae-NK3B31-group* (ρ = −0.32), *Prevotella-1* (ρ = −0.43), *xylanophilum-group* (ρ = −0.39), *Ruminiclostridium-6* (ρ = −0.21), and *Lactobacillus* (ρ = −0.18). In addition, *Prevotellaceae-NK3B31-group* (ρ = 0.27), *xylanophilum-group* (ρ = 0.21), and *Selenomonas* (ρ = 0.23) were positively correlated with weight gain, whereas *Desulfovibrio* (ρ = −0.31), *Ruminococcaceae-UCG-005* (ρ = −0.29), *Parabacteroides* (ρ = −0.18), *Rikenellaceae-RC9-gut_group* (ρ = −0.19), *Christensenellaceae-R.7-group* (ρ = −0.24), *Ruminococcaceae-UCG-010* (ρ = −0.19), and *Family-XIII-AD3011-group* (ρ = −0.28) were negatively correlated with weight gain (Figure 2). Of the above-mentioned bacteria, nine gut microbiota were found to be correlated with both viral load and weight gain (Table 1). Interestingly, these gut microbiota exhibited an opposite correlation with these two traits, suggesting that the gut microbiota have an antagonistic effect on viral loadand weight gain.

### 3.3. Differences in Gut Microbiota between Susceptible and Resistant Pigs

Based on the previous study, PRRSV infection was best controlled by pigs with a constant body weight or a weight gain that inhibits virus replication in the host [39]. The relationship between weight gain and viral load was shown in Figure 3A. The scatter plot showed the four relations of viral load versus weight gain including a high viral load/low weight gain (Hv/Lg), a high viral load/high weight gain (Hv/Hg), a low viral load/low weight gain (Lv/Lg), and a low viral load/high weight gain (Lv/Hg). During PRRSV infection, pigs that survived in the Lv/Hg group were defined as resistant ones to PRRSV, and pigs that died in the Hv/Lg group were designated as susceptible ones to PRRSV. Six pigs with the best and worst clinical signs were selected from resistant pigs and susceptible pigs for subsequent gut microbiota investigations.

To reveal the innate colonization of gut microbiota in resistant and susceptible pigs, we compared the changes in the relative abundance of gut microbiota at genus levels at 0 dpi between two populations. The results indicated that only the relative abundance of *Prevotellaceae-NK3B31**-**group* was significantly lower in resistant pigs than in susceptible pigs, whereas the relative abundance of *Ruminococcaceae-UCG-005*, *Ruminococcaceae-NK4A214-group*, *Ruminococcaceae-UCG-002*, *Campylobacter*, *Lachnospiraceae-XPB1014-group*, *Anaerotruncus*, and *Family-XIII-AD3011-group* was significantly higher in resistant pigs than in susceptible pigs (Figure 3B). Furthermore, the relative abundance of gut microbiota at genus levels was compared between resistant and susceptible populations at 7 dpi. Compared to susceptible pigs, the relative abundance of *Prevotellaceae-NK3B31-group*, *Ruminococcaceae-UCG-014*, and *Ruminococcus-1* was significantly higher in resistant pigs, while the relative abundance of *Campylobacter*, *Methanobrevibacter*, *Desulfovibrio*, and *Family-XIII-AD3011-group* was lower in resistant pigs (Figure 3C). Further exploration indicated that a significant increase in the gut microbiota abundance in the comparison of 0 dpi versus 7 dpi was observed in *Ruminococcaceae-UCG−002*, *Campylobacter*, *Ruminococcaceae-NK4A214-group*, *Ruminococcaceae-UCG-005*, *Anaerotruncus*, *Lachnospiraceae-XPB1014-group*, *and Family-XIII-AD3011-group*. In particular, the *Prevotellaceae-NK3B31-group* was significantly higher in the resistant group and significantly lower in the susceptible group. (Figure 3D).

### 3.4. Function Prediction of Gut Microbiota in Susceptible and Resistant Populations

To explore the relationship between the potential functions of the porcine gut microbiota and PRRS resistance, the gut microbiota functions in the susceptible and resistant populations were analyzed. Based on OTU abundance information and species annotation using Tax4Fun software, 36 level-2 KEGG ORTHOLOGY (KO) groups were obtained. As shown in Figure 4A, only the transcription (adjusted *p* < 0.01) pathway was significantly enriched in the resistant pigs at 0 dpi, whereas 10 KEGG pathways were significantly enriched in both resistant pigs and susceptible pigs at 7 dpi (adjusted *p* < 0.05, Figure 4B). Cell Growth and Death, Endocrine System, Immune System, Cardiovascular Disease, Digestive System, and Carbohydrate Metabolism were significantly enriched in susceptible pigs, while Energy Metabolism, Amino Acid Metabolism, and Xenobiotics Biodegradation and Metabolism were significantly enriched in resistant pigs.

## 4. Discussion

Mammalian gut microbiota are composed of a large number of bacterial, fungal, and other microbial communities [40]. The diversity of the bacterial community provides a variety of biological functions for host immunity [41]. Under normal conditions, the gut microbiota is in a stable and balanced state. When the gut microbiota are attacked by pathogenic bacteria, the balance is broken followed by the change in the gut microbiota community [42]. PRRSV infection causes intestinal mucosal breakdown, hemorrhagic enteritis, diarrhea, and other diseases that are directly related to the immune functions mediated by the gut microbiota. PRRSV suppress host immune responses, thus causing some pathogenic bacteria to multiply, meanwhile, due to the inherent homeostasis regulatory function of the host, pathogen-antagonistic bacteria begin to multiply rapidly, thus, the gut microbiota show an increase in the diversity and abundance after PRRSV infection [43]. In the present study, the Shannon index and Chao1 index were increased upon PRRSV infection, indicating an increase in both gut microbiota diversity and abundance. The fecal microbial composition in PRRSV-infected pigs revealed that *Bacteroidetes*, *Firmicutes*, *Spirochaete*, *Proteobacteria,* and *Chamydiae* were the top five dominant phyla in the pig gut microbiota, which is consistent with the previous findings [44]. The relative abundance of three phyla *Spirochete*, *Proteobacteria*, and *Chamydiae* was increased after PRRSV infection. Two phyla, *Proteobacteria* and *Chamydiae,* have been reported to be closely associated with intestinal inflammation [45], suggesting that an inflammatory response might have occurred after PRRSV infection. In this study, the relative abundance of *Bacteroidetes* and *Firmicutes* was decreased after PRRSV infection, while *Bacteroidetes* and *Firmicutes* were the dominant bacteria in Duroc, Landrace, and Yorkshire pigs, and these two bacteria play an important role in maintaining piglet health [46].

Gut microbiota are involved in the host immune response and play important roles in the host immune homeostasis. *Parabacteroides* has been reported to damage the immune system [47], and is negatively associated with obesity [48]. *Christensenellaceae-R-7-group* exhibits high abundance in bacterially-infected constipated patients, causing a loss of appetite, emotional anxiety, and a negative correlation with daily weight gain [49]. Notably, the genus *Anaerotruncus*, the only one found, has been shown to be associated with bacteremia and cachexia [50]. In this study, we found that viral load was positively correlated with *Parabacteroides* (ρ = 0.44), *Christensenellaceae-R7-group* (ρ = 0.5), and *Anaerotruncus* (ρ = 0.42), and weight gain was negatively correlated with *Parabacteroides* (ρ = −0.18), *Christensenellaceae-R7-group* (ρ = −0.24), and *Anaerotruncus* (ρ = −0.23). This suggests that these bacteria have a positive effect on PRRSV replication and a negative effect on body weight gain. Viral load and weight gain are two important indicators to assess the immune function and growth status of pigs during PRRSVinfection [51], and the impact of PRRSV on the host could be directly demonstrated by these two indicators. This aligned with our results that resistant individuals had a lower viral load and a higher weight gain compared to susceptible individuals. In this study, the correlation of gut microbiota with viral load or weight gain was consistent with the clinical sign manifestation that pigs with severe pathological symptoms gained less weight. Taken together, changes in the gut microbiota of pigs induced by PRRSV infection can affect viral load and body weight.

Understanding the important role of the gut microbiota in response to extraintestinal diseases is an emerging area of attention, such as respiratory viral infections [52]. As shown in our results, gut microbiota and resistance indicators were altered after PRRSV infection. In particular, the microbiota changes in resistant and susceptible individuals caused by PRRSV infection deserve more attention. We found that the species and abundance of beneficial bacteria in the gut were higher in resistant pigs than in susceptible pigs before PRRSV infection. *Ruminococcaceae* promote the production of short-chain fatty acids (SCFAs) in the mammalian gut, and they are responsible for degrading many polysaccharides and fibers, maintaining intestinal barrier function, and preserving the host gut health [53]. In this study, the comparison of gut microbiota between resistant pigs and susceptible pigs before PRRSV infection indicated that the abundance of the Ruminococcaceae-UCG−002, *Ruminococcaceae-NK4A214-group,* and *Ruminococcaceae-UCG−005* were significantly higher in resistant pigs. Interestingly, the probiotic *Prevotellaceae-NK3B31-group* in susceptible pigs had a high abundance before PRRSV infection. However, this bacterium exhibited a significant decrease in abundance with negative increments in susceptible pigs and a positive increment in resistant pigs. The above results suggested that differences in the innate colonization of some microbiota may affect the resistance of the host at a later stage.

The gut microbiota’s response to external stimuli affects the host homeostasis. Swine microbiota will be disrupted afterPRRSVchallenge [54]. The analysis of the gut microbiota in resistant pigs and susceptible pigs after PRRSV infection revealed that the abundance of *Prevotellaceae-NK3B31-group*, *Ruminococcaceae-UCG-014*, and *Ruminococcus-1* was significantly higher in resistant pigs, whereas *Campylobacter*, *Methanobrevibacter*, *Desulfovibrio,* and *Family**-XIII**-AD3011**-group* showed a higher abundance in susceptible pigs. Most dominant bacteria with high relative abundance after PRRSV infection were not dominant before PRRSV infection, except *Prevotellaceae-NK3B31-group*, *Campylobacter,* and *Family-XIII-AD3011-group*. As a probiotic bacterium, *Prevotellaceae-NK3B31-group* can effectively alleviate intestinal inflammatory responses, promote nutrient absorption in the gut, and reduce immune rejection in autoimmune diseases [55], and this bacterium plays a crucial role in utilizing polysaccharides in feeds, thus facilitating the growth of post-weaning piglets [56]. Our results showed that *Prevotellaceae-NK3B31-group* exhibited the greatest change in relative abundance before and after PRRSV infection. *Campylobacter*, an active intestinal pathogen, is considered a potential pathogen of diarrhea [57]. In the study, the relative abundance of gut *Campylobacter* in susceptible pigs was three folds as high as that in resistant pigs at 7 dpi, and the degree of diarrhea was significantly higher in susceptible pigs than in resistant pigs, suggesting that *Campylobacter* might be involved in the pathogenesis of PRRSV diarrhea. *Desulfovibrio*, a Gram-negative bacterium, is considered to be associated with ulcerative enteritis and immune system imbalance, and it is responsible for secreting toxic metabolites [58], while *Methanobrevibacter*, a Gram-positive bacterium, causes abnormal glutamine in the blood, triggering metabolic syndrome [59]. *Family-XIII-AD3011-group*, as a conditionally pathogenic bacterium, causes excessive insulin secretion in gut microbiota dysregulation or other pathogenic bacteria’s excitation states, thus resulting in hypoglycemic symptoms [60] that may be related to depression and metabolic disturbance. Our results show that the gut microbiota in pigs is greatly altered after PRRSV infection, and it is meaningful that the changes in characteristic gut microbiota might be correlated with host resistance.

## 5. Conclusions

This study investigated the changes in gut microbiota of pigs in response to PRRSV infection. We revealed the correlation between viral load or weight gain and gut microbiota and screened the gut microbiota with significant differences from susceptible and resistant pigs. Our results provided insight into the relationship between PRRSV infection and gut microbiota and clarified the biological basis of potential PRRS-resistant traits. The discovered disease resistance-associated strains may provide an effective probiotic resource to prevent and control PRRS.

## Figures and Tables

**Figure 1 animals-12-01504-f001:**
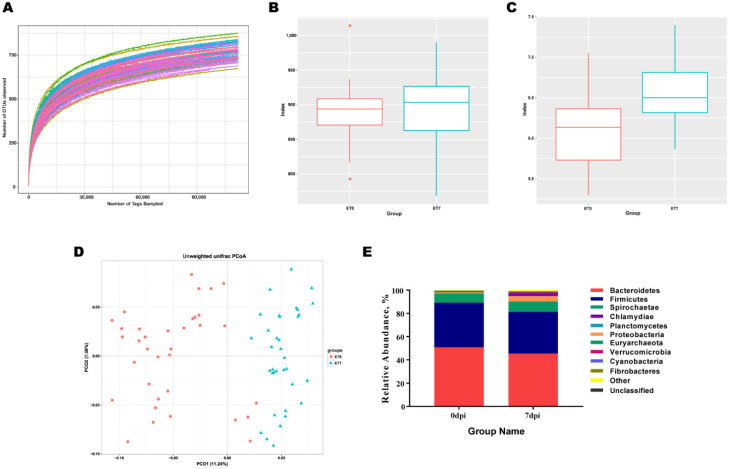
Characterization of sequencing output. (**A**) Rarefaction curves. Each curve represents each pig. (**B**) Chao1 index. (**C**) Shannon index. Different letters above the bars denote a significant difference in alpha diversity index among the groups tested by Wilcoxon signed-rank test and adjusted for false discovery rate (FDR, *p* < 0.05). (**D**) 2D-PCoA plots of gut microbiota of 0 dpi (red circle) or 7 dpi (blue triangle), based on unweighted UniFrac distance in microbial communities. (**E**) Community bar plot analysis of bacterial at the phylum level, only the top 10 species with expression abundance of 2% were displayed, the rest were classified into the other category, and tags that cannot be annotated to this level were classified into the unclassifiedcategory.

**Figure 2 animals-12-01504-f002:**
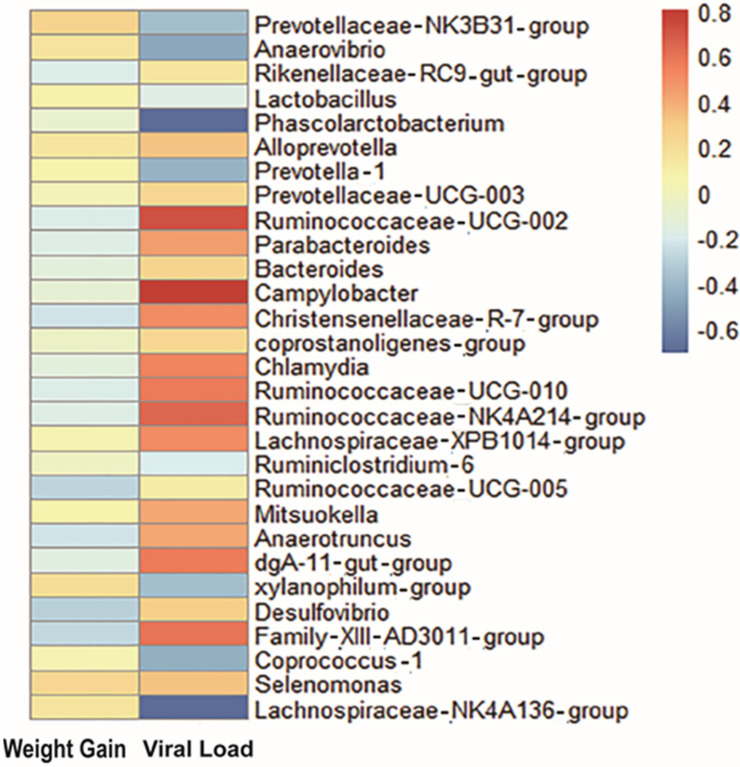
Heatmap analyses of gut microorganisms associated with viral load and weight gain. Spearman’s correlations among the weight gain, viral load, and the microbial genera. The color of the spots in the right panel represents the correlation coefficients (ρ) of the genera with viral loadand weight gain.

**Figure 3 animals-12-01504-f003:**
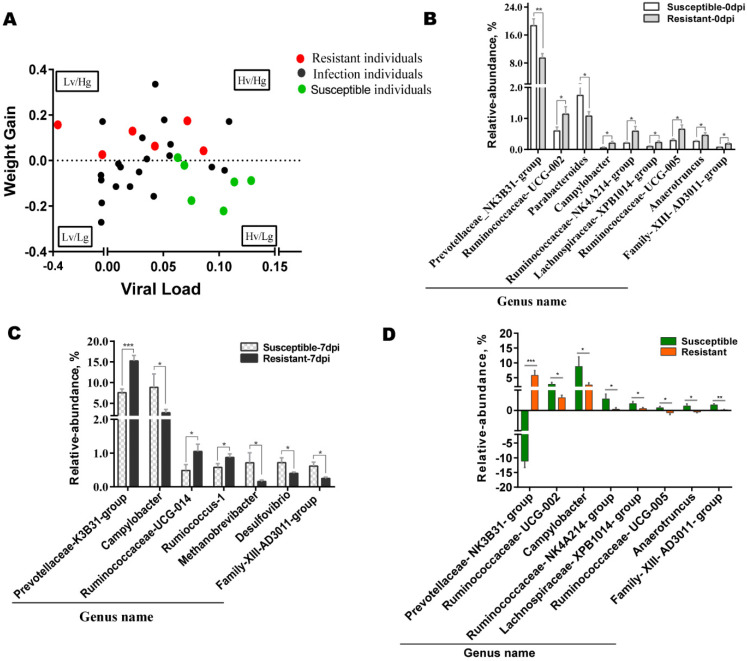
Dynamic change of gut microbiota in susceptible and resistant pigs. (**A**) Viral loadversus weight gain. Hv represents high viral load. Lv represents low viral load. Hg represents high weight gain. Lg represents low weight gain. The red circle represents resistant pigs. The black circle represents experimental pigs. The blue circle represents susceptible pigs. (**B**) Differences of gut microbiota between susceptible and resistant pigs at day 0. (**C**) Differences of gut microbiota between susceptible and resistant pigs at 7 dpi. (**D**) Incremental change in gut microbiota between 7 and 0 dpi. Different letters above the bars denotes significantly differentially abundant genera among groups. (Data are mean ± SE; and * *p* < 0.05; ** *p* < 0.01, *** *p* < 0.001).

**Figure 4 animals-12-01504-f004:**
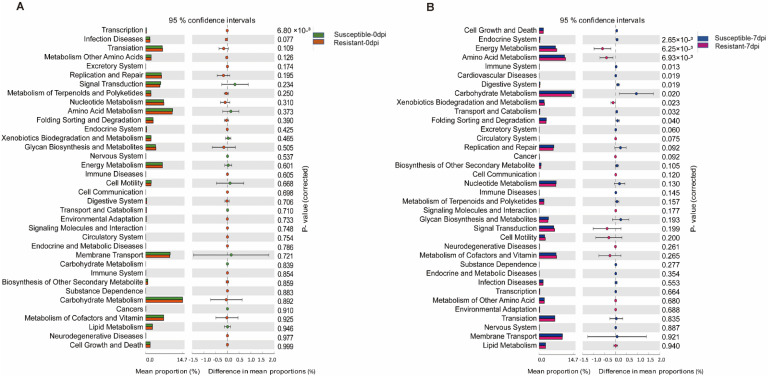
Changes in predicted metagenomic functions of gut bacterial in susceptible and resistant pigs at 0 (**A**) and 7 dpi (**B**), respectively. *p* ≤ 0.05 represents a significant difference. (**A**) green boxes, susceptible 0 dpi group; orange boxes, resistant 0 dpi group; (**B**) blue boxes, susceptible 0 dpi group; red boxes, resistant 7 dpi group.

**Table 1 animals-12-01504-t001:** The co-correlative key gut microbiota between viral load and weight gain.

	Correlation Coefficient	Viral Load	Weight Gain
Genus			
*Prevotellaceae-NK3B31-group*		−0.39	0.27
*Xylanophilum-group*		−0.39	0.21
*Christensenellaceae-R-7-group*		0.50	−0.24
*Parabacteroides*		0.44	−0.18
*Anaerotruncus*		0.42	−0.23
*Family-XIII-AD3011-group*		0.59	−0.28
*Desulfovibrio*		0.29	−0.31
*Ruminococcaceae-UCG-002*		0.71	−0.19
*Ruminococcaceae-UCG-010*		0.57	−0.19

## Data Availability

The manage sequence data that support the findings of this study are submission in the NCBI GenBank at https://www.ncbi.nlm.nih.gov/ (accessed on 7 February 2021) under BioProject: PRJNA803608.

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
