# Peer review of "Change of Gut Microbiota in PRRSV-Resistant Pigs and PRRSV-Susceptible Pigs from Tongcheng Pigs and Large White Pigs Crossed Population upon PRRSV Infection"

_animals, 2022, doi:10.3390/ani12121504_

Round 1
Reviewer 1 Report
The manuscript entitled “Change of Gut Microbiota in PRRSV-resistant pigs and PRRSV-suspectable pigs from Tongcheng pigs and Large White pigs crossed population upon PRRSV infection” described that the resistance of pigs was related to the composition of gut microbiota. Although there is no conclusive results, the authors found that some genus might be related to viremia, clinical symptoms, and disease resistance.
- The word “suspectable” in the title should be replaced by “susceptible”. In the text, there are same mistakes, the authors need check the spelling very carefully. Additionally, there are some mistakes in Singular-Plural Agreement and tense, thus extensively editing of English language must be carried out. As example, “… shows gut microbiota play important role in regulating host immune response to PRRS virus (PRRSV).” “PRRSV Infection induces Increase in Abundance and Biversity of Gut Microbiota”.
- Animal experiment in Materials and Methods section is no clear, the authors should clearly describe the overall experimental design and procedures.
- In the section of results, the authors should clearly present the clinical syndrome, gain weight, survival curve after PRRSV infection.
- In lines 193-195, the virus load was tested, but the results were not showed.
- In table 1, the viral load was confusing. The authors should make it clear.
- In lines 238-239, there were six pigs with the best clinical signs. But, in figure 3A, there were only five individual dots.
- In lines 379-380, the tense of the sentence needs corrected.
Reviewer 2 Report
The study authored by Tengfei et al., investigated the changes of the gut microbiota in the susceptible and resistant pigs upon PRRSV infection. The manuscript was well written by the authors and explained well in the methods and the discussion part. The introduction part very clear. Since, there are few corrections and clarifications are needed for the improvement of the article.
Major corrections
1. Titile: Suspectable pigs or susceptible pigs?, Follow the similar terminology as in the main text. Suspectabe is not meaningfull.
2.Line 113-114: Is the any additional result for justifying the sentence about the negative test results on selected pigs? if, provide the data in the supplementary files.
3. The authors must change the word <viral load> to viral replication in the figure 2. is it not seems that the Viral load like viral injection to the individuals?
4. Line 200-215: The repeated info similar to Figure 2. Inform any important microbial population in the text. The authors must provide the numerical values of weight gain and the viral replication (viral load) for the microorganisms as a table in the main text or supplementary files.
5. Line 212: Since, it is confusing, which one is for weight gain and viral load? two values on the same organism: Ruminococcaceae_UGC-005?
6. Line 217-219: Not only 9 gut bacteria, other few species are also showed an opposite correlation with these two traits (Ex. Phascolactobacterium; Anaerovibrio; lachnospiraceae). Verigy it.
7. Figure 2. Does the weight gain and the viral load are propotionaly inversed?
8. What are the PRRSV resistance species in the Figure 2.
9. Lines 245-246: Missing Lachnospiraceae? verify it according to the sentence context.
10. Line 255: Prevotellaceae_NK3B31_group showed negative abundance? but the sentence showed increased in the gut microbiota? correct it.
Minor corrections
1. Lines (104, 193, 236, 271, 284, 359): more space between letters or after full stops.
2. Similarly, lines (119, 312, 325, 335): No space between letters or after full stops. correct as single space accordingly.
3. Line 165: Write the 70 fecal samples info in the methodology part.
4. Lines 178-179: Write properly, comma separated genus names.
5. Table-1: correction for -039 to (-0.39) for prevotellcea_NK3B31.
6. Figure 4. The figure a,b are not visually clear. Enlarge the figures.
Reviewer 3 Report
The article "Change of Gut Microbiota in PRRSV-resistant pigs and PRRSV-suspectable pigs from Tongcheng pigs and Large White pigs crossed population upon PRRSV infection" is written very well. The results are interesting. The paper has enriched the global knowledge about the correlation between gut-microbiota and the persistence to PRRSV. I have notice only small errors such as:
Line 28: should be space between words „population.PRRS”
Line 111: please add country where the farm was located
Line 121: please add country of the university
Line 195: previous was used “ml” now “mL”, please be consequent
Line 273: “P” is written in italic, in previous parts of article was not, please be consequent
Line 341: should be space between words “pigs.Interesting”
Except that the article is very good and should be published after minor revision.
Sincerely,
Reviewer
Round 2
Reviewer 1 Report
No further comments.
Reviewer 2 Report
The authors have provided the corrections and modifications in the main text. Appreciate for the improvement of the article.